# Moderating Effects of Religious Tourism Activities on Environmental Risk, Leisure Satisfaction, Physical and Mental Health and Well-Being among the Elderly in the Context of COVID-19

**DOI:** 10.3390/ijerph192114419

**Published:** 2022-11-03

**Authors:** Hsiao-Hsien Lin, Tzu-Yun Lin, Chun-Wei Hsu, Che-Hsiu Chen, Qi-Yuan Li, Po-Hsuan Wu

**Affiliations:** 1Department of Leisure Industry Management, National Chin-Yi University of Technology, Taichung 41170, Taiwan; 2Department of Sport Information and Communication, National Taiwan University of Sport, Taichung 404401, Taiwan; 3College of History, Culture and Tourism, Yulin Normal University, Yulin 537000, China; 4Department of Sport Performance, National Taiwan University of Sport, Taichung 404401, Taiwan; 5School of Physical Education, Jiaying University, Meizhou 514015, China; 6Department of Environmental Science and Engineering, National Pingtung University of Science and Technology, Pingtung 912301, Taiwan

**Keywords:** Baishatun, Mazu, religious and cultural tourism, the elderly, risks in tourism, well-being

## Abstract

The purpose of this study is to explore whether religious tourism activities can create a safe leisure environment and improve the well-being of the elderly during the COVID-19 pandemic, with the participants in the Baishatun Mazu pilgrimage in Taiwan as the subjects of this study. A mixed research method was used. First, statistical software and the Pearson product-moment correlation coefficient were used to analyze the data. Then the respondents’ opinions were collected. Finally, a multivariate analysis method was used to discuss the results of analysis. The findings showed that the elderly respondents thought that the epidemic prevention information and leisure space planning for the pilgrimage made them feel secure. The elderly believed the scenery, religious atmosphere, and commodities en route could reduce the perception of environmental risks to tourists, relieve pressure on the brain, and increase social opportunities. Therefore, the friendlier the leisure environment around the pilgrimage, the greater the leisure satisfaction among the elderly respondents. The happier the elderly felt, the less they considered the concentration of airborne contaminants, including viruses. The better their physical and mental health was, the less likely they were to want to ask for religious goods.

## 1. Introduction

Due to the advancement of science and technology and the quality improvement in healthcare, the public health has been generally improved [1]. In addition, countries around the world have made remarkable progress in economic development, people’s living standards have improved, and social welfare has increasingly improved [2]. These have led to a reduction in infection rates and even fatality rates, and an increase in average life expectancy. However, with fluctuations in the global economy, outbreaks of epidemics, wars, and other crises, the cost of living, consumption, and care has increased, and people’s willingness to have children has declined [3]. As a result, the birth rate has been declining year by year, leading to a population imbalance [3,4]; an aging society has become a challenge that many countries are currently facing [5,6].

An aging society is defined as a country or region in which the share of the population aged over 65 exceeds seven percent of the whole population [7]. An aging social and population structure have positive and negative influences on national development [6,8]. Because people’s knowledge, skills, and abilities increase as they get older, and they have more experience, friends, and resources [9], meaningful life experiences and techniques can be passed on when there are more elderly people. The elders of a society are both the inheritance of human civilization and wisdom and one of the most influential human resources in urban development [10]. However, as people grow older, their self-healing ability and immunity decline, and the elderly are more vulnerable to viruses [11]. In addition, the elderly have been the most affected by severe COVID-19 infections, and had strikingly higher COVID-19 mortality rates compared to younger individuals [12]. When the elderly suffer from COVID-19, they have a harder time bouncing back, and use far more health care services than do younger groups [13,14], affecting their quality of life, their physical, and mental health, and increasing the social and economic burdens on the country [9,15]. Therefore, ways in which societies can create a safe living environment for the elderly, improve their quality of life, help them maintain good health, reduce the consumption of healthcare resources, and stabilize the national economic development are the main focus of this study.

Culture consists of symbols and activities such as writing, literature, art, customs, and religious culture. It is a systemic sum of symbols or sounds commonly recognized by human beings in the process of continuous evolution, a phenomenon, and a manifestation of mentality or habit [16]. Religion is an obsession that occurs when people worship a person, thing, or object that is natural or transcendent and satisfies their physical or psychological needs [16,17]. It tends to make individuals passionate about this culture, which leads to unshakeable beliefs and eventually becomes a common belief of individuals or communities [17] or a special cultural phenomenon [18]. Religion is supernatural, and the power of worship and belief cannot be explained [19]. In addition, these religious beliefs, which are derived from the natural or transcendent environment, have further developed into myths [16] and shaped the places of their origin into religious sanctuaries [18]. With the evolution of time, these religious sites and historical relics have become tourist sites with historical, cultural, and tourism values [20]. Thus, religion is a kind of representation of human civilization with local cultural and historical significance and rich in leisure and tourism resources [21]. It can be a motivation for people to engage in leisure or tourism activities, sightseeing, site visit, worship, study, and meditation [22], and has the functions of boosting the confidence of the masses, reducing their anxiety [23], and achieving healthy social development [24,25]. We believe that the elderly should be able to achieve the goal of stabilizing the mind, maintaining good physical and mental health, reorganizing daily routines, and improving quality of life by participating in religious activities. In addition, local governments can use these activities to boost the confidence of the masses, restore social order, and stimulate economic development.

Baishatun Mazu, whose full name is Tianshang Shengmu of Baishatun Gongtian Temple, is one of the main Goddesses worshiped by local people in Taiwan. Baishatun Gongtian Temple, located in Miaoli County in western Taiwan, was founded in 1863 and has a history spanning more than 200 years [26]. Although it has experienced the hazards of typhoons, earthquakes, wars, and aging, the appearance of the temple and Taoist culture have been well preserved and maintained by believers [26,27]. In its early days, the belief was aimed at gathering the strength of the believers, establishing a belief in love and gratitude, and creating a safe living environment [26]. The pilgrimage was planned and made every year from Miaoli County to Yunlin County, which lasts eight days and seven nights, over a total of about 400 km [27]. After years of evolution, it has become a cultural tourism activity in which people can regulate their emotions, gather group cohesiveness, and deepen their religious beliefs by getting close to nature and the local religious and cultural environment [28]. This activity has a history covering more than a hundred years, and is a unique intangible cultural asset in Taiwan. It is a religious tourist activity and has an international reputation [27]. Although there have long been pre-arranged accommodations for the pilgrims, in fact these have often been randomly selected en route [29]. In addition, organizers might provide the registered participants with identification items such as shutter caps, souvenir t-shirts, armbands, etc., as well as cultural and creative products such as incense bags and charms [30]. Along the way, local believers spontaneously prepared food and drinking water for participants to sample for free [28]. In addition, in order to reduce the risk of virus infection, the organizer required that participants needed to receive three doses of vaccine in advance, wear masks during the pilgrimage, do their best to avoid conversations, maintain physical distancing during meals, eat separate meals, etc. [31], in order to provide participants with a safe leisure environment and help them gain a good travel experience. According to statistics, more than 100,000 people participated in the whole process of pilgrimage during the COVID-19 pandemic in 2022 [32]. It is obvious that this activity remained deeply trusted by the people during the pandemic, as it brought the benefits of leisure to the participants and satisfied their emotional needs.

Scholars believe that because most organs function less well as people age, the costs of health maintenance for the elderly are high [13,14]; under the influence of the COVID-19 pandemic the costs of maintaining the physical and mental health of the elderly are rising even higher [12]. The Baishatun Mazu pilgrimage is a religious tourist activity [27] in which people can regulate their emotions, gather group cohesiveness, and deepen their religious beliefs [28]. It is both a leisure and a tourist activity [29,30,31,32]. Therefore, we believe that during the current COVID-19 pandemic, the environmental risk perception of elderly people can be decreased by participating in religious and cultural activities in which they have a short-term safe living space, gain sufficient leisure benefits, and achieve a sense of happiness. In addition, when environmental risk perception is lower, leisure satisfaction is higher [33]. The more powerful the perception of good physical and mental health is on the part of the elderly [34], the happier they will feel [35,36].

On the other hand, studies have shown that there are differences between the expected effects of decision-making and the actual effects [20,21,37,38,39]. Different environments and human or non-human factors may cause this difference [21,38]. Furthermore, it takes time to solve these questions, and only participants with practical experiences and perceptions are able to provide the right answers [20,39]. As such, we believe we can acquire the answer by collecting data from the perceptions of those elderly participants who have actually made the pilgrimage. In addition, the main purposes of previous studies on elderly participation in religious activities or cultural tourism have been explorations of religious culture [40] and economic, social, and environmental benefits [41,42]. Most of them have been conducted to investigate the influence on people’s quality of life [43] and physical health [44] after the pilgrimage. Although a few studies have been carried out to investigate issues such as satisfaction and well-being [45], none were aimed at analyzing of the relationship between leisure satisfaction, physical and mental health, and well-being, let alone the exploration of the influence of religious and cultural tourist activities on the environmental risk perception, leisure satisfaction, physical and mental health, and well-being of elderly participants.

In summary, religious activities can enhance positive emotions, and safe travel decision planning can shape a safe environment. Therefore, we hypothesized that these factors should lead to lower environmental risk perceptions, higher leisure satisfaction, better physical and mental health, and higher well-being among the elderly after participating in religious tourism activities. However, we found that such studies are currently lacking, and have not been discussed in the context of the COVID-19 pandemic. Therefore, we considered that discussing the issue of religious and cultural tourism on environmental risk, leisure satisfaction, physical and mental health, and well-being of the elderly in the pandemic environment would be meaningful and is a current research gap that deserves discussion. Therefore, the purpose of our study was to confirm the effects of religious and cultural tourism on the environmental risk perception, leisure satisfaction, physical and mental health, and well-being of the elderly in the pandemic environment. We believe that the results of this manuscript can help governments to take effective, safe, and rational measures to promote good health to the elderly. In addition, suggestions are offered to help the elderly make good preparations to obtain benefits, stay in a positive frame of mind, preserve their health, and live a happy live. Hopefully, this will be the main significance and contribution of this manuscript.

## 2. Literature Discussion

### 2.1. Environmental Risk to Tourists

Risk refers to the possibility that something bad, unpleasant, or dangerous may happen [46]. When people participate in tourist activities, they may have doubts about the space or environment in which they stay. They may think that there may be unpredictable or even more serious potential problems during the travel [47]. We call this environmental risk to tourists.

The causes of risk are often at odds with everyday knowledge, and are predictable [48]. A tourist’s reaction to the environmental risk is his/her perception of the environmental risk [49]. Scholars believe that risks are related to uncertainty, possible losses, and future occurrences [50,51]. When an individual has a perception of uncertainty about a thing or behavior, the thing or behavior in question is likely to have a negative impact or even lead to a bad result [51]; the stronger this perception of uncertainty is, the greater the risk [52]. Predicting visitors’ environmental risk perceptions can be discussed through issues such as the natural environment, air quality, event atmosphere, commodities, routes, precautions, and resting spaces [50,51,52,53]. Therefore, we believe that the environmental risk perception of elderly tourists can be assessed based on the natural environment, air quality, activity atmosphere, commodities, routes, preventive measures, and rest spaces.

On the other hand, although religious tourist activities provide the benefits of soothing emotions, relieving stress, and promoting health [28], the symptoms of COVID-19 are unclear, the incubation period of the virus is long, and it spreads fast [12]. In addition, activities attract people and increase physical contact [31], contributing to the fast spread of the virus. In particular, the elderly have a higher morbidity and mortality rate [13,14]. All of these factors may increase the environmental risk perception of elderly tourists [9,13,14,15]. Therefore, we believe that when the elderly participate in religious and cultural tourist activities during the COVID-19 pandemic, they should have a negative environmental risk perception. As a result, we assume that all the older adults have a negative environmental risk perception as tourists, which is Research Hypothesis 1.

### 2.2. Leisure Satisfaction

Satisfaction refers to one’s pleasure response when a system meets their desire and wants [54]. Leisure satisfaction refers to the positive perceptions or feelings that an individual forms, elicits, and gains as a result of engaging in leisure activities and choices [51].

Leisure satisfaction is the use of personal activity experience to compare previous experience, personal expectations, or sense of expectation [54]. Scholars believe that when the actual leisure environment or activity content meets personal expectations, a satisfying attitude emerges [55]. With a higher the satisfaction level, higher positive energy cognition can be obtained [56], along with greater satisfaction of personal psychological needs [57]. The influencing factors of leisure satisfaction can be divided into internal and external interference [58], which can be analyzed from such differing perspectives as physiology, psychology, education, aesthetics, social interaction, accommodation, and relaxation [59]. Further discussion can take place based on the aspects of mass confidence, cultural identity, supplies, activity design, accommodation, epidemic prevention measures, recreational space, and the environment [42,58]. Therefore, we believe that the leisure satisfaction perceived by elderly tourists can be assessed based on issues such as mass confidence, cultural identity, supplies, activity design, accommodation, epidemic prevention measures, and the recreational environment.

The Baishatun Mazu pilgrimage is a 400 km round trip from Baishatun to Mioli, and most participants complete the pilgrimage on foot [27]. Participants can rest at will; the organizer provides shuttle bus services, and there are medical staff and rest areas available en route [29,30]. Therefore, the participants should have access to both a safe leisure environment and adequate exercise during the pilgrimage. As a result, we believe that when the elderly participate in religious and cultural tourist activities during the COVID-19 pandemic, they should have a positive perception of leisure satisfaction. Therefore, we assume that all the elderly participants have a positive perception of leisure satisfaction, which is Research Hypothesis 2.

### 2.3. Physical and Mental Health

When a person feels good and pleasant about his or her physical, psychological, and social conditions, that means the person is in good physical and mental health [58]. This represents an analytical method of self-perception evaluation [59,60], and can present the actual situation through scientific test evidence such as self-assessment [57]. The higher a person’s health risk is, the greater its influence on individual behavioral decisions and perceptions [12,61,62].

Physical and mental health cognition is usually assessed based on personal feelings and the influence of the current environment on a person’s physical and mental health [12]. Scholars believe that higher perception of positive physical and mental health is related to a person’s better adaptability in their environment [63], which means that their environment is a safe space that fits the conditions for people to live or rest [64]. Physical and mental health can be assessed based on three levels: psychology, spirit, and attitude [58,62,65]. Currently, headache, insomnia, stomach pain, and anxiety are common problems [66]. Therefore, we believe that the perception of physical and mental health on the part of the elderly can be assessed based on their headache, insomnia, stomach pain, anxiety, and other aspects.

Religious activities can provide the elderly with the health-promoting benefits of calming the mind and relieving stress [28], enabling individuals to have a good experience in physical, psychological, and social interactions and to achieve positive perception of their physical and mental health [58]. Therefore, we believe that when the elderly participate in religious and cultural tourist activities during the COVID-19 pandemic, they should have a positive perception of their physical and mental health. On this basis, we assume that the elderly participants have a positive perception of their physical and mental health, which is Research Hypothesis 3.

### 2.4. Well-Being

Well-being is achieved on the basis of the positive development of individuals, which is a complex and subjective psychological attitude [67]. It is measured based on personal subjective emotions and feelings and preferences [68]; when material needs are satisfied and the interactive atmosphere in a person’s surroundings is recognized [69], this is happiness.

The results of earlier studies have shown that well-being can be measured based on economic indicators such as GDP [70], while more recent studies have indicated that it is necessary to include the individual’s health and welfare status as well [71] in order for these to be considered as complete well-being indicators. Personal perception of well-being can be judged by indicators such as whole-body relaxation, life experience, life planning, and social interaction [72]. Therefore, we believe that the well-being of the elderly can be assessed based on their physical and mental relaxation, enjoyment of life, fulfilling life planning, and active interpersonal interaction.

Religious tourist activities have the benefits of calming people’s minds, relieving stress, and promoting health [28]. Scholars believe that religious activities can help the elderly to have their souls comforted and gain life sustenance [73]. This helps to stabilize the mood of the elderly and allow them to obtain a good quality of life [20]. Therefore, we believe that the elderly should have a positive perception of well-being when participating in religious and cultural tourist activities during the COVID-19 pandemic. As a result, we assume that the elderly all have positive feelings in their perception of well-being, which is Research Hypothesis 4.

### 2.5. Influence of Tourists’ Environmental Risk Perception on Leisure Satisfaction, Physical and Mental Health, and Well-Being

Baishatun Mazu pilgrimage is a long journey with diverse scenery, and can provide sufficient exercise and leisure space to offer opportunities for the elderly to obtain leisure benefits [60], maintain their physical and mental health [58,62,65], and achieve well-being [70,71,72]. However, due to the unstable health status of many elderly people, along with low energy and relatively weaker immune systems, prolonged exposure to sun and walking may be harmful to them [61]. In addition, the vaccination rates are low in Taiwan; the elderly are less willing to be vaccinated, a large number of people gather during the pilgrimage [31], they are at high risk from COVID-19 [12], the elderly are more likely to become very sick from COVID-19, and they have high mortality rate [13,14]. For all of these reasons, we believe that when religious and cultural activities involve many environmental risks to tourists during the COVID-19 pandemic, the elderly may be less willing to participate in activities, spend less time on relaxing activities, and obtain less satisfaction from participating in activities. In addition, in an unsafe environment the elderly cannot find leisure benefits, relieve stress, or solve their physical and mental health problems. As a result, elderly participants may not be able to gain a safe and happy experience.

Therefore, based on the above inferences, we believe that the environmental risk perceptions of elderly tourists participating in religious and cultural tourist activities during the COVID-19 pandemic will interfere with leisure satisfaction, physical and mental health, and well-being. Therefore, Hypothesis 5-1 is that the environmental risk perceived by tourists has a significant impact on their leisure satisfaction; Hypothesis 5-2 is that the environmental risk perceived by tourists has a significant impact on their physical and mental health; and Hypothesis 5-3 is that the environmental risk perceived by tourists has a significant impact on their well-being.

## 3. Methods

### 3.1. Framework and Hypothesis

This study was designed to take the Baishatun Mazu pilgrimage in Taiwan as an example in order to analyze whether religious and cultural tourist activities can create a friendly leisure environment and promote the physical and mental health of the elderly based on views collected from the elderly participants in the activity. A mixed research method was adopted to measure the participants’ environmental risk perception, leisure satisfaction, and physical and mental health. The research framework is shown in Figure 1.

Based on a literature analysis and inference results, we propose five hypotheses.

**Hypothesis** **1.**
*Elderly tourists have negative environmental risk perception when participating in the Baishatun Mazu pilgrimage during the COVID-19 pandemic.*


**Hypothesis** **2.**
*Elderly tourists have positive leisure satisfaction perception when participating in the Baishatun Mazu pilgrimage during the COVID-19 pandemic.*


**Hypothesis** **3.**
*Elderly tourists have positive mental and physical health perception when participating in the Baishatun Mazu pilgrimage during the COVID-19 pandemic.*


**Hypothesis** **4.**
*Elderly tourists have positive perception of their well-being when participating in the Baishatun Mazu pilgrimage during the COVID-19 pandemic.*


**Hypothesis** **5.**
*The environmental risk perceived by tourists has a significant impact on leisure satisfaction (5-1); the environmental risk perceived by tourists has a significant impact on their perceived physical and mental health (5-2); the environmental risk perceived by tourists has a significant impact on their perceived well-being (5-3).*


### 3.2. Process and Tools

This study aims to confirm whether religious and cultural tourism activities can be used by the elderly to mitigate perception of environmental risk, maintain leisure satisfaction, and regulate physical and mental health and well-being in the COVID-19 pandemic environment. As very little research has been published investigating this topic, a quantitative study could improve or the breadth [73], followed by a qualitative study to increase the depth [74], while a multivariate review analysis could compensate for any deficiencies in the applied theories or research methods [75].

When studying the literature related to the Baishatun Mazu pilgrimage, elder health, environmental risks to tourists, leisure satisfaction, and physical and mental health and well-being, we found that the relevant research topics and issues have not been much discussed by scholars. Therefore, we decided to make up for the lack of related research topics, methods, or theories though mixed research methods [76]. As a result, quantitative methods were adopted to broaden the scope of the research [74,77]. Then, a questionnaire was edited according to the literature on environmental risks to tourists [46,47,48,49,50], leisure satisfaction [51,52,53,54,55,56,57,58], physical and mental health [59,60,61,62,63,64,65,66], and well-being [67,68,69,70]. Four experts in tourist decision-making, travel management, public health, and science of religion were commissioned to check the content validity. After completing the preliminary draft, 100 questionnaires were collected in March 2022, and SPSS 26.0 statistical software was used to conduct reliability analysis. According to the analysis results, α coefficients of the total scale of environmental risks to tourists, leisure satisfaction, physical and mental health, and well-being all exceeded 0.8, indicating that the reliability of all items was good and acceptable for further use and analysis [78]. After the questionnaire was confirmed, 800 questionnaires were distributed from 22–27 May 2022, of which 720 questionnaires (90%) were returned. After excluding incomplete questionnaires, 599 valid questionnaires (83.2%) were obtained. The questionnaire data were analyzed by basic statistical tests and the PPMCC test.

When the results of the questionnaire analysis had been obtained, qualitative research was used to increase the depth of the research [73]. Semi-structured interviews were conducted and twelve respondents were invited to provide their opinions on the results of the questionnaire analysis. Finally, we compiled the data from multiple sources and discussed the data from various perspectives [79,80,81,82], hoping to obtain more accurate answers for judgment. Expert background, questionnaire reliability analysis, interviewee background, and overviews of the issues are shown in Table 1.

### 3.3. Methods and Limitations

Mixed methods were used for this study. However, due to the threat of the COVID-19 pandemic coupled with restrictions on funds, manpower, and material resources, the researchers simultaneously commissioned the surveyors to conduct field surveys, used a convenience sampling method for sampling, sent the questionnaire online to the elderly participants in the 2022 Baishatun Mazu pilgrimage platform, and used a snowball sampling method to expand the sample. In addition, we invited experts in the fields of tourist decision-making, travel management, public health, and science of religion, as well as event organizers, staff members, temple administrators, tour guides, and elderly participants, to help us determine the interview survey questions. Then, we contacted the respondents by video phone and telephone to ask them about their willingness to be interviewed. Next, we conducted the survey in the form of semi-structured interviews and asked the respondents to put forward their opinions on the questionnaire analysis and results. Finally, all the data were collected and discussed by means of multivariate examination.

However, as explained above, due to the impact of the risk of infection during the survey and the limitations of the sampling methods, research methods, and the willingness of participants to be interviewed, the results of this study were not perfect. Thus, we provide several suggestions for improvement and look forward to future investigations that may implement them.

### 3.4. Ethical Considerations

The survey was conducted anonymously. All research tools were standardized and compiled through rigorous processes. During the survey, the research assistants explained the research topic, purpose, participation methods and questions to the respondents, reaffirmed their willingness to participate, and obtained authorization to use the survey data. Then, the respondents’ views were presented anonymously. Finally, after summarizing, classifying, and comparing the data, we analyzed the data from multiple perspectives. All surveys and sampling processes were designed in accordance with the principles of fairness, openness, and impartiality [44,83], with the principle design of Taiwan Executive Yuan Health Bureau Bulletin No. 1010265075 [84], and in conformity with Articles 1004 and 1009 of the Civil Code of the People’s Republic of China [85].

## 4. Analysis and Discussion

Analysis was conducted based on data collected from 599 valid questionnaires. We found that among all respondents’ backgrounds, 356 (59.4%) were male and 243 (40.6%) were female, 554 (92.5%) were aged 61–70, and 45 (7.5%) were aged over 71. It is obvious from these data that elderly men were more willing to participate and that people over 71 years old were less willing to participate.

### 4.1. Environment Risks to Elderly Tourists, Their Leisure Satisfaction, Physical and Mental Health and Well-Being

As shown in Table 2, a basic statistical test was used to analyze the participants’ perception of environmental risk to tourists, leisure satisfaction, physical and mental health, and well-being. It was found that in terms of the perceptions of environmental risk to tourists, issues such as beautiful scenery, religious atmosphere, and activity-related goods (3.62) were the strongest, and the traffic flow plan (3.47) was the weakest. In terms of the perceptions of leisure satisfaction, the perceptions of strict anti-epidemic measures and secure rest spaces (3.66) were the strongest, and the perception of smooth activity design (3.58) was the weakest. In terms of physical and mental health perception, the perception of relieving headaches (2.54) was the strongest, and the perception of relieving stomach pain (2.31) was the weakest. In terms of well-being, the perception of interpersonal relationships (2.86) was the strongest, and the perception of relaxation (2.55) was the weakest.

According to the above analysis, the scenery along the way, religious atmosphere, and activity-related goods reduced the participants’ perception of environmental risk during the pilgrimage. In addition, the leisure environment and epidemic prevention measures were able to improve leisure satisfaction, relieve headaches, and improve the interpersonal relationships among the elderly participants. However, the traffic flow plan might lead to the environmental risks to tourists, gastrointestinal disorders, anxiety, and other problems. Therefore, it is obvious that there is no consistency in the elderly participants’ perceptions of environmental risk, leisure satisfaction, physical and mental health, and well-being, showing that Hypotheses 1 to 4 are disproved.

### 4.2. Correlation Analysis of Environmental Risks to Tourists, Leisure Satisfaction, Physical and Mental Health, and Well-Being

As shown in Table 3, Pearson product-moment correlation analysis was used to analyze the correlation between environmental risks to tourists, leisure satisfaction, physical and mental health, and well-being. We found a significant correlation between environmental risks to tourists and leisure satisfaction, physical and mental health, and well-being (*p* < 0.01). Therefore, it is obvious that the respondents’ perception of environmental risks to tourists has a significant influence on their leisure satisfaction, physical and mental health, and well-being. These results show that research Hypotheses 5-1, 5-2, and 5-3 are proved.

Among these, leisure space has a positive and significant impact on leisure satisfaction, activity-related goods have a negative and significant impact on physical and mental health, and open air has a negative and significant impact on well-being. In addition, religious atmosphere has a significant influence on group cohesiveness, activity-related goods on cultural identity and the accommodation environment, traffic flow planning on supplies, and rest space on supplies, traffic flow plan, epidemic prevention measures, and rest space. Activity-related goods have a negative and significant effect on relieving headaches, insomnia, and anxiety. Open and fresh air has a negative and significant impact on current life, enthusiasm for life, activity-related goods, and relaxation.

### 4.3. Discussion

#### 4.3.1. Perception of Environmental Risks to Tourists, Leisure Satisfaction, Physical and Mental Health, and Well-Being on the Part of Elderly Participants in Religious and Cultural Tourist Activities

We believe that religion helps to spread culture, customs, and knowledge and to promote education, recreation, travel, meditation, and worship. In addition, when people participate in religious and cultural activities or purchase activity-related goods they are often engaged in gathering group cohesiveness, exercising, reducing their anxiety, and promoting their physical and mental health. Moreover, the cities and townships the participants in the Baishatun Mazu pilgrimage pass through are different from each other in terms of urban development, local characteristics, temples, and surrounding scenery. Therefore, when the elderly participate in religious tourist activities, the scenery, religious atmosphere, and activity-related goods along the way can make them forget about environmental risks to tourists. This phenomenon is consistent with the results in the literature [17,18,19,21,58,62]. However, uncertainties involve risks such as epidemic infection and accidents during tourist activities. The number of participants in the pilgrimage was large, the route changes could occur any time, and the participants were different from each other in terms of the anti-epidemic measures they took. Furthermore, bigger crowds suggest that the virus is able to spread more easily. As a result, the traffic flow plan for the pilgrimage was not able to reduce the risk perception of the participants. This phenomenon is inconsistent with the results from the literature [17,18,19,21,58,62].

Second, due to the long-term promotion of public health education in Taiwan, the public has a strong awareness of self-health monitoring and a sincere willingness to follow the epidemic prevention policy. In addition, the government has experience in the prevention and control of SARS viruses, and the COVID-19 epidemic continues to spread. The organizer has offered a sufficiently spacious rest area, according to the instructions of the CDC, for participants to maintain social distance while resting and relaxing. Therefore, when the elderly respondents participated in the religious activity, they gained sufficient benefits of leisure activities due to the rest environment and epidemic prevention measures during the pilgrimage. This phenomenon is consistent with the results in the literature [17,18,19,21,58,62,63]. However, because the virus spreads fast and the Baishatun Mazu pilgrimage route change could occur any time, the huge crowd of participants could not be sure that they would be able to rest in the places arranged by the organizer. These uncertainties during the pilgrimage might have reduced the leisure satisfaction of the elderly. This phenomenon is inconsistent with the results from the literature [17,18,19,21,58,62,63].

Next, because religion plays a prominent role in defining the culture of a society and Baishatun Mazu is one of the main beliefs of the Taiwanese people, it promotes education, leisure activities, tourism, and entertainment, puts people’s minds at ease, and relieves their stress. Therefore, when the elderly respondents participated in religious tourist activities, they felt relaxed and enjoyed relief from headaches. This phenomenon is consistent with the results in the literature [17,18,19,21,58,62,63,86]. However, the COVID-19 pandemic situation was severe, the virus spreads fast, there are a large number of chronically ill patients in Taiwan, overall vaccination rate was low, the intentions of the elderly to get vaccines were weak, and overall immunity to the virus was insufficient. In addition, there were a large number of people participating in the pilgrimage, increasing the risk of infection, the Baishatun Mazu pilgrimage procession moved quickly, and the traffic flow changed greatly [87]. All this led to irregular meal times for the participants during the pilgrimage, making them tend to suffer from anxiety and stomach pain. This phenomenon is inconsistent with the results from the literature [17,18,19,21,58,62,63,86].

In addition, because the epidemic has not yet eased, the country’s economic development has been affected, and the public felt on edge about the pandemic and needed to calm themselves down by doing something. Many people in Taiwan like to participate in religious activities, and the Baishatun Mazu pilgrimage is one of the religious activities loved by Taiwanese believers. The temple has over a hundred years of history, and the pilgrimage procession has been held for decades. It represents the inheritance of Taiwanese culture, conveys the concept of gratitude, and has the effect of appeasing believers. The pilgrimage procession provides an opportunity to participate in activities and to promote interaction, friendly communication, and the good social morals of mutual cooperation and mutual trust. Therefore, elderly participants in religious tourist activities had greater opportunities to interact with other believers, obtain health information, and enhance their personal and social relationship with relatives and friends. This phenomenon is consistent with the results in the literature [17,18,19,21,58,62,63,86]. However, due to the rapid transmission of the virus, the procession route change could occur any time and the crowds were prone to close contact, increasing the risk of infection; thus, the participants could not forget their worries and feel completely relaxed. As a result, the respondents could not feel totally relaxed during the pilgrimage and gain more happiness. This phenomenon is inconsistent with the results from the literature [17,18,19,21,58,62,63,86].

#### 4.3.2. The Influence of Elderly Respondents’ Perception of Environmental Risks to Tourists on Leisure Satisfaction, Physical and Mental Health, and Well-Being

We believe that the elderly are prone to be anxious about crowded places where transmission of COVID-19 may spread more easily. In addition, if people do not have enough time and space for exercise, physiological functions decline faster and their energy levels decrease more rapidly, seriously affecting health. However, religion has the effect of purifying the soul, extending belief, and gathering group cohesiveness, and tourist activities have the effects of recreation, entertainment, and education. Moreover, there were beautiful scenery, commodities, and landscapes that the participants could enjoy en route, and the organizer and the public reached a consensus on epidemic prevention and control measures. Therefore, the elderly respondents had confidence in the pilgrimage, their perception of the environmental risks to tourists decreased, and their leisure satisfaction increased. As a result, we believe that when environmental risks to tourists are low, the result is high leisure satisfaction; for elderly participants, good rest spaces are vital. This phenomenon is consistent with the results in the literature [17,18,19,21,58,62,63,86].

Due to the long-term impact of the COVID-19 pandemic, the elderly had far fewer opportunities for leisure activities and lacked enthusiasm for life, resulting in anxiety and a sense of helplessness. Religion can have the effect of educating and appeasing believers, while tourism has the effects of recreation, rest, and relaxation. At that time, the elderly participating in religious tourist activities could find opportunities for leisure, entertainment, and both physical and mental relaxation. In addition, the Taiwanese government has long promoted public health education and epidemic prevention awareness, and the elderly have high health literacy and awareness with respect to following the epidemic prevention policy. Therefore, the elderly respondents felt that they would participate in religious tourist activities as long as they thought they were in good health, happy, thankful, and had good thoughts, even if they knew that route change might occur any time, the crowd would be huge, they could not keep to physical distancing, and the air quality would not be good. They expected to take the opportunity to relieve their anxiety and stress, gain spiritual comfort and relaxation, build interpersonal relationships with others, and improve their health and well-being. Therefore, the elderly respondents were willing to participate in activities even if the activities were risky as long as they attained opportunities for leisure activities and the improvement of their physical and mental health and well-being. We believe that this is one of the reasons that strong perceptions of environmental risks have a negative impact on physical and mental health and well-being which activity-related goods and air quality do not. This phenomenon is inconsistent with the results from the literature [17,18,19,21,58,62,63,86].

Finally, because of the influence religious and tourist activities have on people, the related cultural and creative and tourism products are valuable and meaningful to them. In addition, the organizers have strengthened epidemic prevention measures and continuously promoted the epidemic prevention policy during the event to achieve the effect of enhancing the public’s awareness of personal hygiene, epidemic prevention, and cooperation. Moreover, there were various food supplies, rest spaces and emergency medical facilities available en route to improve safety and comfort on the pilgrimage. Therefore, when the elderly respondents believed that the religious atmosphere, activity-related goods, traffic flow plan, rest spaces, and epidemic prevention measures were all in place, their positive perception of cultural identity, consensus-building, accommodation environment, and supplies was strong. Therefore, there is a positive correlation between religious atmosphere, activity-related goods, traffic flow planning, rest spaces, and epidemic prevention measures on the one hand and cultural identity, group cohesiveness, accommodation environment, and food supplies on the other. This phenomenon is consistent with the results in the literature [17,18,19,21,58,62,63,86]. However, because the constant route changes and crowded gatherings increased the chance of contact and infection, the elderly respondents believed their headache, insomnia, and anxiety would not be relieved, and they could not attain sufficient relaxation or arouse their enthusiasm and confidence for life without activity-related goods and high air quality. Therefore, activity-related goods and air quality are negatively correlated with headache, insomnia, anxiety, enthusiasm for life, and self-confidence. This phenomenon is inconsistent with the results from the literature [17,18,19,21,58,62,63,86].

## 5. Conclusions

We found that religious and cultural tourism activities have educational, leisure, and recreational functions as well as stabilizing, stress-relieving, and consensus-building functions. Therefore, such activities can indeed regulate the environmental risk awareness on the part of the elderly, increase their leisure satisfaction, improve physical and mental health, and increase well-being in the pandemic environment. Moreover, when participants have high awareness of epidemic prevention and trust in epidemic prevention management decisions, it can create a safe activity environment, promote leisure benefits for the elderly, reduce the perception of environmental risks, relieve stress, and improve headache problems. Furthermore, the better the physical and mental health and sense of well-being of the elderly, the less they care about goods, air circulation, and route planning, and the more they want to participate in leisure activities in order to improve their physical and mental health and sense of well-being. With better open space planning and experience perception, a higher satisfaction level can be achieved with respect to leisure activities. We believe that if the changes in movement can be reduced and extensive disinfection and cleaning can be carried out before transfers, it is possible to divert the flow of people and reduce the gathering time. In addition, strengthening personal immunization measures and increasing vaccination rates can enhance virus resistance, thereby reducing the risk of environmental infections, increasing leisure satisfaction, and preventing irregular eating, anxiety, and stomach pains, ultimately improving the physical and mental health of the elderly and enhancing their sense of well-being.

Based on the results of the above analysis and discussion, we provide the following suggestions to government, organizers, the public, etc.

For the government:

The governments should increase COVID-19 tests and vaccination stations and develop qualified medical products, increase the number of medical supplies and reduce testing costs, and encourage the public to get vaccinated.

2.For organizers and the public:

Institutions should increase publicity and manpower for epidemic prevention and control, and strengthen participants’ awareness and literacy with respect to epidemic prevention. The elderly should follow the epidemic prevention policy of the government and the organizer, follow their doctor’s assessment and advice, and get vaccinated as soon as possible to enhance individual and herd immunity.

3.For follow-up research development:

As explained above, due to the impact of the risk of infection during the survey and the limitations of the sampling methods, research methods, and willingness of participants to be interviewed, the results of this study were not perfect. Therefore, we suggest that future research be conducted looking into differences between countries, age groups, and frequency of vaccination. We may investigate differences in the perceptions of participants in other religious cultures and between fixed and non-fixed religious practices. We believe that if the above recommendations are completed, it can help to improve the weaknesses of the present study.

## Figures and Tables

**Figure 1 ijerph-19-14419-f001:**
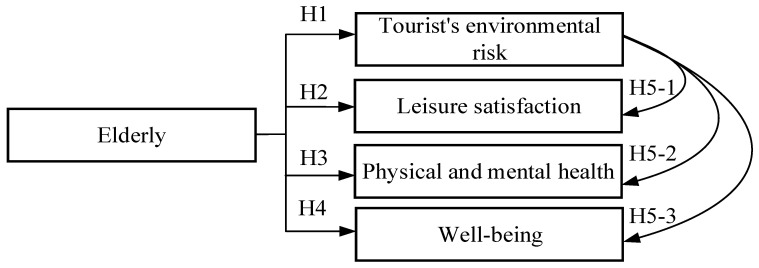
Study framework.

**Table 1 ijerph-19-14419-t001:** Expert background, questionnaire reliability analysis, interviewee background, and overviews of the issues.

Facet (α)	Issue			α	KMO	Bartlett (χ^2^)	df	*p*
Tourist’s environmental risk (0.878)	The beautiful scenery made me forget about the risk of infection (beautiful scenery);			0.861	0.914	1747.971	21	0.000
	Fresh and open air made me forget about risk of infection (fresh and open air)			0.855				
	The sacred religious atmosphere made me forget about the risk of infection (religious atmosphere)			0.834				
	Goods related to religious culture and activity made me forget the risk of infection (activity-related goods)			0.865				
	The traffic flow plan of the pilgrimage made me forget about the risk of infection (traffic flow plan)			0.872				
	Epidemic prevention measures for the pilgrimage made me forget about the risk of infection (epidemic prevention planning)			0.870				
	The rest space and environment of the pilgrimage made me forget about the risk of infection (rest space)			0.876				
Leisure satisfaction (0.887)	The pilgrimage could gather group cohesiveness (gather group cohesiveness)			0.869	0.921	1873.594	21	0.000
	The pilgrimage could enhance cultural identity (enhance cultural identity)			0.872				
	Satisfied with the supplies for the pilgrimage (satisfied with the supplies)			0.874				
	Smooth event design for the pilgrimage (smooth event design)			0.872				
	Good accommodation for the pilgrimage (good accommodation)			0.870				
	Strict anti-epidemic measures for the pilgrimage (strict anti-epidemic measures)			0.872				
	The rest space and environment for the pilgrimage were secured (secured rest space)			0.875				
Physical and mental health (0.953)	Participating in the pilgrimage relieved headaches (relieve headaches)			0.915	0.818	2940.182	6	0.000
	Participating in the pilgrimage relieved insomnia (relieve insomnia)			0.938				
	Participating in the pilgrimage relieved stomach pain and increase appetite (relieve stomach pain and increase appetite)			0.942				
	Participating in the pilgrimage relieved anxiety (relieve anxiety)			0.952				
Well-being (0.907)	Participating in the pilgrimage helped me to relax (relax)			0.886	0.824	1717.021	6	0.000
	Participating in the pilgrimage made my current life happy (current life happy)			0.864				
	Participating in the pilgrimage made me feel meaningful (feel meaningful)			0.890				
	Participating in the pilgrimage allowed me have more opportunities to interact with others (good interpersonal relationship)			0.905				
	gender	specialty	age	gender		specialty		age
Professionals or interviewees	male	tourist decision-making	42	female		public health		58
	male	travel management	48	female		science of religion		52
	male	event organizer	52	male		temple administrator		65
	male	staff member	49	female		Tour guide		65
	female	staff member	58	female		elder		67
	male	temple administrator	66	male		elder		63

**Table 2 ijerph-19-14419-t002:** Analysis of the respondents’ perceptions of environmental risks to tourists, leisure satisfaction, physical and mental health, and well-being.

Facet	Issue	SD	M	Rank	Dimension	Issue	SD	M	Rank
Tourist’s environmental risk	Beautiful scenery	1.181	3.62	1	Physical and mental health	Relieve headache	0.849	2.54	1
	Fresh and open air	1.135	3.58	2		Relieve insomnia	0.854	2.38	3
	Religious atmosphere	1.163	3.62	1		Relieve stomach pain and increase appetite	0.852	2.31	4
	Activity-related goods	1.172	3.62	1		Relieve anxiety	0.925	2.45	2
	Traffic flow plan	1.142	3.47	4	Well-being	Relax	0.904	2.55	4
	Epidemic prevention planning	1.113	3.58	2		Current life happy	0.925	2.65	2
	Rest space	1.199	3.57	3		Feel meaningful	0.945	2.56	3
Leisure satisfaction	Gather group cohesiveness	1.184	3.61	3		Good interpersonal relationship	0.990	2.86	1
	Enhance cultural identity	1.110	3.61	3					
	Satisfied with the supplies	1.210	3.64	2					
	Smooth event design	1.149	3.58	5					
	Good accommodation	1.186	3.59	4					
	Anti-epidemic information & measures	1.192	3.66	1					
	Secured rest space	1.133	3.66	1					

**Table 3 ijerph-19-14419-t003:** Correlation analysis of environmental risks to tourists, leisure satisfaction, physical and mental health, and well-being.

Facet	Perception of Environmental Risks to Tourists	Beautiful Scenery	Open and Fresh Air	Religious Atmosphere	Activity-Related Goods	Traffic Flow Plan	Epidemic Prevention Plan	Rest Space
Leisure satisfaction	0.883 **	0.682 **	0.660 **	0.677 **	0.712 **	0.641 **	0.594 **	0.725 **
Gather group cohesiveness		0.566 **	0.532 **	0.568 **	0.537 **	0.545 **	0.470 **	0.553 **
Enhance cultural identity		0.461 **	0.503 **	0.481 **	0.576 **	0.450 **	0.425 **	0.518 **
Satisfied with the supplies		0.535 **	0.532 **	0.527 **	0.547 **	0.577 **	0.535 **	0.577 **
Smooth event design		0.517 **	0.509 **	0.520 **	0.530 **	0.463 **	0.409 **	0.550 **
Good accommodation		0.571 **	0.544 **	0.534 **	0.574 **	0.488 **	0.466 **	0.538 **
Anti-epidemic information & measures		0.552 **	0.469 **	0.525 **	0.567 **	0.467 **	0.445 **	0.615 **
Secured rest space		0.478 **	0.477 **	0.500 **	0.516 **	0.470 **	0.458 **	0.566 **
Physical and mental health	−0.066 *	−0.040	−0.056	−0.024	−0.101 *	−0.079	0.007	−0.051
Relieve headache		−0.080 *	−0.102 *	−0.084 *	−0.143 **	−0.101 *	−0.030	−0.127 **
Relieve insomnia		−0.023	−0.036	−0.010	−0.083 *	−0.067	0.020	−0.026
Relieve stomach pain and increase appetite		−0.009	−0.010	0.001	−0.071	−0.068	0.020	0.000
Relieve anxiety		−0.037	−0.061	0.000	−0.085 *	−0.061	0.016	−0.041
Well-being	−0.192 **	−0.113 **	−0.187 **	−0.169 **	−0.155 **	−0.144 **	−0.083 *	−0.167 **
Relax		−0.069	−0.115 **	−0.086 *	−0.150 **	−0.131 **	−0.054	−0.126 **
Current life happy		−0.135 **	−0.214 **	−0.182 **	−0.161 **	−0.139 **	−0.082 *	−0.159 **
Feel meaningful		−0.137 **	−0.187 **	−0.161 **	−0.175 **	−0.126 **	−0.081 *	−0.183 **
Good interpersonal relationship		−0.059	−0.145 **	−0.166 **	−0.068	−0.116 **	−0.075	−0.124 **

* *p* < 0.05, ** *p* < 0.01.

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
