# Peer review of "Moderating Effects of Religious Tourism Activities on Environmental Risk, Leisure Satisfaction, Physical and Mental Health and Well-Being among the Elderly in the Context of COVID-19"

_ijerph, 2022, doi:10.3390/ijerph192114419_

Round 1
Reviewer 1 Report
“Can Religious Tourism” is a well-researched and insightful paper. I have only minor comments.
In line 26, To break up an overly long sentence and correct a minor grammatical error, I recommend “secure. The elderly also believed the scenery…” instead of “secured, as well as the scenery…”
In line 45, “a population” and “an aging society” should replace Population” and “the aging society”
In line 47, “An aging society” and “a country” should replace “The aging society” and “the country”
In line 51, the word “excellent” is a questionable word choice insofar as it raises questions about what excellence is and who determines it – “meaningful” or “meaningful and valuable” would be better because they suggest that what you are discussing is meaningful and/or valuable to the persons in question.
In line 52, to break up an overly long sentence, I recommend “people. The elders of a society are both” instead of “people both”
In line 56, “infection” should be “infections”
In line 58 “Covid-19” should be “COVID-19” and all other instances of “Covid-19” in the paper should be changed to “COVID-19”
In line 61, for greater clarity, I suggest “how societies can create” rather than “how to create”
In lines 64-78, the discussions of culture, civilization, and religion could be more clearly presented and more precise. Religion is more than just belief; it is a particular kind of belief. Specifically, in is belief in ultimate reality. All religious beliefs are about transcendent or ultimate meaning and significance, but many religious beliefs, including the beliefs of the major religions of the world, cannot be said to be purely supernatural. There are naturalistic beliefs, that is, beliefs about finding the holy or the ultimate in nature, in most religions, and there are naturalistic religions. Additionally, rather than writing that religious mythologies become religious shrines and sites, it would be more accurate to say that religious mythologies are embodied or represented in religious shrines and sites.
In lines 84 and 85 it is not clear what the authors means when they say “the belief was aimed” – what belief are they talking about here? Do they mean the pilgrimage to the Baishatun Mazu temple?
In line 135-136, for greater clarity, I recommend “to discuss the environmental risk” rather than, “to discuss issues such as environmental risk” and then later in the sentence I recommend “satisfaction, and physical” rather than “satisfaction, physical”
In line 142, I recommend ”promote the good health of the elderly” rather than “promote good health to the elderly”
In line 153, “and unpredictable” should be “and are unpredictable”
In lines 156-157, I recommend “about some thing or behavior, it will likely” rather than “about something or behavior, the thing or behavior performed will likely”
The sentence that runs from lines 159-161 is unclear, and need to be rewritten.
In line 226, why include the word “career”? Well-being is achieved by positive personal development on multiple levels, not just career development.
In line 450, “SARS virus” should be “SARS viruses”
In line 502, “few” should be “fewer”
The writing in lines 575-578 is somewhat awkward and unclear. I suggest the authors rewrite it to try to express more clearly and fully the point they want to make.
Author Response
Reviewer 1
“Can Religious Tourism” is a well-researched and insightful paper. I have only minor comments.
We are very grateful to the reviewers for their willingness to commit Chen to review and make good-natured suggestions to improve this manuscript, and thank you for your endorsement of the manuscript.
We will fix the review recommendations and provide responses based on each question. (Relevant amendments such as manuscript red font)
In line 26, To break up an overly long sentence and correct a minor grammatical error, I recommend “secure. The elderly also believed the scenery…” instead of “secured, as well as the scenery…”
Thanks to the reviewer suggestion, we have rewritten this paragraph.
In line 45, “a population” and “an aging society” should replace Population” and “the aging society”
Thanks to the reviewer's suggestion, we have rewritten this paragraph.
In line 47, “An aging society” and “a country” should replace “The aging society” and “the country”
Thanks to the reviewer's suggestion, we have rewritten this paragraph.
In line 51, the word “excellent” is a questionable word choice insofar as it raises questions about what excellence is and who determines it – “meaningful” or “meaningful and valuable” would be better because they suggest that what you are discussing is meaningful and/or valuable to the persons in question.
Thanks to the reviewer's suggestion, we'll use the word "meaningful" instead.
In line 52, to break up an overly long sentence, I recommend “people. The elders of a society are both” instead of “people both”
Thanks to the reviewer for the suggestion, we have adopted your suggestion and have changed to “people. The elders of a society are both…” .
In line 56, “infection” should be “infections”
Thanks for the reviewer suggestion, we adopted your suggestion and used "infections" instead
In line 58 “Covid-19” should be “COVID-19” and all other instances of “COVID-19” in the paper should be changed to “COVID-19”
Thanks to the reviewer for the suggestion, this is indeed a spelling error and we have corrected all mistypes.
In line 61, for greater clarity, I suggest “how societies can create” rather than “how to create”
Thanks to the reviewer for the suggestion, we have adopted your suggestion and have changed to "how societies can create"
In lines 64-78, the discussions of culture, civilization, and religion could be more clearly presented and more precise. Religion is more than just belief; it is a particular kind of belief. Specifically, in is belief in ultimate reality. All religious beliefs are about transcendent or ultimate meaning and significance, but many religious beliefs, including the beliefs of the major religions of the world, cannot be said to be purely supernatural. There are naturalistic beliefs, that is, beliefs about finding the holy or the ultimate in nature, in most religions, and there are naturalistic religions. Additionally, rather than writing that religious mythologies become religious shrines and sites, it would be more accurate to say that religious mythologies are embodied or represented in religious shrines and sites.
Thanks to the reviewer suggestion, we will rewrite the narrative of this paragraph. Strengthen the definitions of culture and religion, and articulate their relevance to the subject of this article.
In lines 84 and 85 it is not clear what the authors means when they say “the belief was aimed” – what belief are they talking about here? Do they mean the pilgrimage to the Baishatun Mazu temple?
Thanks to the reviewer for the question, since the Baishatun Mazu circumnavigation is a mobile religious and cultural activity, it also has a tourist nature. It is held regularly every year.
Most people will participate in this activity, and at a fixed time, choose their appropriate distance to participate in this trip.
Therefore, the "pilgrimage" in our article refers to the public participation in "travel activities".
In line 135-136, for greater clarity, I recommend “to discuss the environmental risk” rather than, “to discuss issues such as environmental risk” and then later in the sentence I recommend “satisfaction, and physical” rather than “satisfaction, physical”
Thanks to the reviewer's suggestion, we have completely rewritten this paragraph. “In summary, although this study also explores the impact of the COVID-19 virus on humans….”
In line 142, I recommend ”promote the good health of the elderly” rather than “promote good health to the elderly”
Thanks to the reviewer's suggestion, we have completely rewritten this paragraph. “In summary, although this study also explores the impact of the COVID-19 virus on humans….”
In line 153, “and unpredictable” should be “and are unpredictable”
Thanks to the reviewer suggestion, we have fixed "and are predictable".
In lines 156-157, I recommend “about some thing or behavior, it will likely” rather than “about something or behavior, the thing or behavior performed will likely”
Thanks to the reviewer suggestion, we have corrected “When an individual has a perception of uncertainty about some thing or behavior, it will likely…”
The sentence that runs from lines 159-161 is unclear, and need to be rewritten.
Thanks to the reviewer suggestion, we have rewritten this paragraph.
In line 226, why include the word “career”? Well-being is achieved by positive personal development on multiple levels, not just career development.
Thanks to the reviewer suggestion, we have removed "career"
In line 450, “SARS virus” should be “SARS viruses”
Thanks to the reviewer's suggestion, we have fixed "SARS viruses"
In line 502, “few” should be “fewer”
Thanks to the reviewer suggestion, we have fixed "fewer"
The writing in lines 575-578 is somewhat awkward and unclear. I suggest the authors rewrite it to try to express more clearly and fully the point they want to make.
Thanks to the reviewer's suggestion, we have rewritten this paragraph.

Reviewer 2 Report
The article is interesting, but it is it has a very chaotic structure, inadequately prepared technically, and is missing very important, structured parts that should be included in a research paper.
- the title of the article is too long;
- research gap and aim of the study should be underlined in the Introduction section;
- the Literature review section should contain information on how other researchers measured safety in tourism, risk perception and other examined elements of tourism;
- method section, please justify why the indicated methods were chosen;
- limitations should be at the end of the article;
- theoretical and managerial implications are missing in conclusions;
- in discussions, it is necessary to add information which results are similar to the results of previous studies and what is different.
- technical underdevelopment in lines 69-70, 186-187, in tables and others.
Author Response
Reviewer 2
The article is interesting, but it is it has a very chaotic structure, inadequately prepared technically, and is missing very important, structured parts that should be included in a research paper.
We are very grateful to the reviewers for their willingness to commit Chen to review and make good-natured suggestions to improve this manuscript, and thank you for your endorsement of the manuscript.
We will fix the review recommendations and provide responses based on each question. (Relevant amendments such as manuscript red font)
the title of the article is too long;
Thanks for the suggestion, we have reworked the header content.
- research gap and aim of the study should be underlined in the Introduction section;
Thanks to the suggestion, we have strengthened the research gap and purpose in the introduction. As in the manuscript, the last paragraph of the Introduction ” In summary, although this study….“
- the Literature review section should contain information on how other researchers measured safety in tourism, risk perception and other examined elements of tourism;
Thanks for the suggestion, we have information on important check elements such as travel safety in our literature review.例如:
Environmental Risk to Tourists:
Therefore, we believe that the environmental risk perception of the elderly tourists can be assessed based on the natural environment, air quality, activity atmosphere, commodities, routes, preventive measures, and rest spaces.
Leisure Satisfaction:
Therefore, we believe that the leisure satisfaction perceived by the elderly can be assessed based on issues such as mass confidence, cultural identity, supplies, activity design, accommodation, epidemic prevention measures, and recreational environment.
Physical and mental health:
Therefore, we believe that the perception of physical and mental health of the elderly can be assessed based on their headache, insomnia, stomach pain, anxiety and other aspects.
Well-being:
Therefore, we believe that the well-being of the elderly can be assessed based on their physical and mental relaxation, enjoyment of life, fulfilling life planning, and active interpersonal interaction.
- method section, please justify why the indicated methods were chosen;
Thanks to the suggestion, we supplement the description of the specified research method in Section 3.2. ” This is a report on confirming that the elderly are in the epidemic environment… ”
- limitations should be at the end of the article;
Thanks for the suggestion, we have adjusted the study limitation description to the end of the article.
- theoretical and managerial implications are missing in conclusions;
Thanks to the suggestion, we have rewritten the content of the conclusions and strengthened the theory and management measures.
- in discussions, it is necessary to add information which results are similar to the results of previous studies and what is different.
Thanks for the suggestion, in the Discussion section, we have added a literature comparison at the end of each discussion's narrative.
- technical underdevelopment in lines 69-70, 186-187, in tables and others.
Thanks to the suggestions, we have improved the narrative of lines 69-70; 186-187, as well as the table presentation technique.

Round 2
Reviewer 2 Report
Thank you for improved version of your paper. But still it needs a lot of corrections.
Please think about theory proposal. How to explain what did you find?
The aim of the study and the research gap do not correspond with title and results.
Conclusions are pure.
Author Response
Thank you for your affirmation of the revised manuscript. We have revised the manuscript again based on the recommendations of this review and responded to questions. The revised content is shown in red font in the manuscript, and the relevant instructions are as follows.
Please think about theory proposal. How to explain what did you find?Thanks for the suggestion, we have stated the relevance of the theory and the issue in the statement of the theory.
For example: "Therefore, we believe that when the elderly participate in religious and cultural tourist activities during the COVID-19 pandemic, they should have a negative environmental risk perception. "
The aim of the study and the research gap do not correspond with title and results.
Thanks for the suggestion, we have adjusted the title to match the purpose, discussion, and conclusion.
Conclusions are pure.
Thanks to the suggestion, we adjusted the narrative of the conclusion.
We really appreciate your once again assisting with the review and making very good suggestions.
We look forward to your approval of this revised manuscript.